# Intrafibrillar Dispersion of Cuprous Oxide (Cu_2_O) Nanoflowers within Cotton Cellulose Fabrics for Permanent Antibacterial, Antifungal and Antiviral Activity

**DOI:** 10.3390/molecules27227706

**Published:** 2022-11-09

**Authors:** Matthew B. Hillyer, Sunghyun Nam, Brian D. Condon

**Affiliations:** Cotton Chemistry and Utilization Research Unit, Southern Regional Research Center, Agricultural Research Service, United States Department of Agriculture, New Orleans, LA 70124, USA

**Keywords:** cuprous oxide, copper, nanoparticles, nanotechnology, cotton, textiles, antimicrobial, antibacterial, antifungal, antiviral

## Abstract

With increasingly frequent highly infectious global pandemics, the textile industry has responded by developing commercial fabric products by incorporating antibacterial metal oxide nanoparticles, particularly copper oxide in cleaning products and personal care items including antimicrobial wipes, hospital gowns and masks. Current methods use a surface adsorption method to functionalize nanomaterials to fibers. However, this results in poor durability and decreased antimicrobial activity after consecutive launderings. In this study, cuprous oxide nanoparticles with nanoflower morphology (Cu_2_O nanoflowers) are synthesized in situ within the cotton fiber under mild conditions and without added chemical reducing agents from a copper (II) precursor with an average maximal Feret diameter of 72.0 ± 51.8 nm and concentration of 17,489 ± 15 mg/kg. Analysis of the Cu_2_O NF-infused cotton fiber cross-section by transmission electron microscopy (TEM) confirmed the internal formation, and X-ray photoelectron spectroscopy (XPS) confirmed the copper (I) reduced oxidation state. An exponential correlation (R^2^ = 0.9979) between the UV-vis surface plasmon resonance (SPR) intensity at 320 nm of the Cu_2_O NFs and the concentration of copper in cotton was determined. The laundering durability of the Cu_2_O NF-cotton fabric was investigated, and the superior nanoparticle-leach resistance was observed, with the fabrics releasing only 19% of copper after 50 home laundering cycles. The internally immobilized Cu_2_O NFs within the cotton fiber exhibited continuing antibacterial activity (≥99.995%) against *K. pneumoniae*, *E. coli* and *S. aureus*), complete antifungal activity (100%) against A. niger and antiviral activity (≥90%) against Human coronavirus, strain 229E, even after 50 laundering cycles.

## 1. Introduction

Metal and metal oxide nanomaterials exhibit remarkable physical, chemical and optical properties due to their nanoscopic size and high surface area-to-volume ratio that are unique and different than their corresponding bulk materials. These include silver, gold, palladium, platinum, chromium, ruthenium, manganese, zinc, nickel, zirconium, cobalt, iron, and copper [1]. Particularly, the versatile and cost-effectiveness of copper has guided researchers toward investigating the application of copper and its oxide nanomaterials in catalysis [2,3], gas-phase sensing [4,5], electrochemistry [6], energy production and storage [7,8], and pharmaceuticals [9,10]. Due to the utility of these nanomaterials they have also found use in commercial products such as agrochemicals, cosmetics, paints, foods, medical devices, and antimicrobial agents [11,12,13]. The textile industry frequently employs the copper and copper oxide (cupric and cuprous) nanoparticles (Cu_x_ONPs, where x = 1 or 2) to imbue dye fastness, UV-protection, and antibacterial activity to cotton fibers [14,15].

While synthetic fibers have become popular in recent years, there has been a market trend towards increased use of cotton for commercial textiles. Laundering and disposal of synthetic fibers lead to the release of microplastics into water supplies and ecosystems [16]. Many recent studies have revealed the ecotoxicological impacts these microplastics have had on human health [17], and on fish [18], plants [19], and microorganisms in aquatic ecosystems [20]. Cotton is an eco-friendly and sustainable alternative to synthetic fibers. Additionally, cotton is durable, a great insulator, and retains moisture far more effectively than synthetic fibers, finding utility in many personal and medical textile products [21]. However, due to its hygroscopicity cotton fibers are susceptible to damage by microorganisms, which can lead to myriad of negative health effects in humans including skin irritation or infections [22,23]. For this reason, copper and copper oxide (cupric and cuprous) nanomaterials have been used as antimicrobial coatings, imparting resistance to microorganism growth. Antibacterial experiments comparing the oxidation state of copper suggest that metallic copper (0)- and copper (I)-containing nanomaterials are the most microbicidal, followed by copper (II) [24].

There are many methods available for synthesizing Cu_x_ONPs, such as sol-gel, laser ablation, sonochemical, electrochemical reduction, coprecipitation, biological, microwave irradiation, etc. [25,26,27]. Many of these synthetic methods are simple and high yielding, however, they require toxic reducing agents and are energy demanding [28]. Cu_x_ONPs are prepared and subsequently applied to cotton fibers by either dip-pad-dry or by using chemical binding agents [29,30,31]. To this end, the externally applied Cu_x_ONPs can be easily detached during use and laundering, which has led to concerns regarding the potential environmental impacts [32]. These impacts can include toxigenic contamination of ground and municipal water supplies, agricultural soils, and bioaccumulation in aquatic ecosystems [11,33,34]. Here, this work presents the in situ synthesis of copper (I) oxide nanoflowers (Cu_2_O NFs) from a copper (II) precursor without using a chemical reducing or binding agent under mild conditions, which results in the internal incorporation of Cu_2_O NFs dispersed within the cotton fiber. A variety of techniques including UV-vis spectroscopy, X-ray diffraction (XRD), X-ray photoelectron spectroscopy (XPS), energy-dispersive X-ray (EDS) spectroscopy, transmission electron microscopy (TEM) and field emission scanning electron microscopy (FE-SEM) were used to determine the morphology and composition of Cu_2_O NF produced within cotton fiber. The washing durability and copper release behavior were investigated using a simulated accelerated home laundering procedure and copper content determined by inductively coupled plasma-mass spectrometry (ICP-MS) and UV-vis spectroscopy. Finally, the antibacterial, antifungal, and antiviral activity of the Cu_2_O NF-cotton fabric were studied.

## 2. Materials and Methods

### 2.1. Materials

Bleached and desized cotton print cloth fabric was purchased from Testfabrics, Inc. (West Pittston, PA, USA). Copper sulfate anhydrous (CuSO_4_, ≥99.99%), sodium hydroxide solution (NaOH, 50% w/v), methyl methacrylate, butyl methacrylate, methyl ethyl ketone, and nitric acid solution (HNO_3_, 68–70% w/w) were purchased from Sigma-Aldrich (St. Louis, MO, USA). All chemicals were used without further purification. Deionized (DI) water was used for the synthesis of Cu_2_O NFs and washing experiments. Milli-Q water (18.0 MΩ/cm) was used in the acid digestion of cuprous oxide-treated cotton fabrics.

### 2.2. In Situ Synthesis of Cu_2_O NF-Cotton Nanocomposite Fiber

To 80 mL of DI water was dissolved 0.250 g cupric sulfate anhydrous. To the stirring solution of copper sulfate was slowly added 20 mL of 50% w/v sodium hydroxide solution to produce a 0.250 g·100 mL^−1^ stock solution. Upon addition of sodium hydroxide, the clear and light blue solution (CuSO_4_) initially precipitated a light blue solid (Cu(OH)_2_), which dissolved upon complete addition of the 20 mL of 50% w/v sodium hydroxide solution to become a clear dark blue solution ([Cu(OH)_4_]^2−^). Additional solutions were prepared from this 0.250 g·100 mL^−1^ stock solution by dilution with 10% w/v sodium hydroxide solution with concentrations 0.250 g·100 mL^−1^, 0.100 g·100 mL^−1^, 0.050 g·100 mL^−1^, 0.010 g·100 mL^−1^, 0.005 g·100 mL^−1^, 0.001 g/100 mL^−1^, and a control sample with only sodium hydroxide. A 50 mm × 200 mm swatch of bleached and desized cotton print cloth fabric was added to a 50 mL centrifuge tube and 40 mL of [Cu(OH)_4_]^2−^ solution was added. The centrifuge tube was mixed at 750 rpm for 30 min using an Advanced Vortex Mixer (Fisher Scientific, Hampton, NH, USA). The fabric changed from a white to dark blue color. The fabric was then removed from the solution and rinsed with DI water to remove excess copper precursor and sodium hydroxide, changing from dark blue to light blue when the excess base was removed. The fabric was added to 80 °C DI water, where the light blue fabric changed immediately to dark brown. The fabric was removed, rinsed in DI water and air dried at room temperature.

### 2.3. Cross-Sectional Analysis of Cu_2_O NF-Cotton Nanocomposite Fiber

To observe internal dispersion of Cu_2_O NFs within the cotton fiber, a procedure developed at the Southern Regional Research Center was used [35,36]. Fibers from the treated cotton fabric were combed and immersed in a methacrylate matrix solution in Teflon tubing (3.2 mm inner diameter), which was polymerized using UV light for 30 min. The resulting block of encased fiber was removed from the Teflon tubing and immobilized in polyethylene capsules. Using a PowerTome Ultramicrotome (Boeckeler Instruments, Inc., Tucson, AZ, USA), the fibers were sliced into 100–120 nm sections for TEM. For TEM analysis, the sections were placed on a carbon-film-coated copper grid. The polymer matrix was then dissolved using methyl ethyl ketone.

### 2.4. Characterization of Cu_2_O NF-Cotton Nanocomposite Fiber

Transmission electron microscopy (TEM) was used to image Cu_2_O NFs prepared inside cotton fiber by a TEM (FEI Tecnai G2 F30) operating at 300 kV. Field emission scanning electron microscopy (FE-SEM) equipped with EDS was used to examine external morphology and surface composition of Cu_2_O NF-cotton fibers using an SEM (JSM-6610 LV, JEOL) operating with an acceleration voltage of 5.0 keV. Samples for FE-SEM analysis were prepared with a carbon coating. Cuprous oxide nanoflower sizes were measured using TEM images by counting particles with ImageJ software (NIH) [37]. Size and distribution were determined from a representative sample of 924 particles.

X-ray diffractograms (XRD) of the cotton fabric fibers were collected using a PANalytical Empyrean X-ray Diffractometer (Malvern Panalytical, Malvern, UK) with Cu Kα radiation (1.54060 Å) and generator settings of 45 kV and 40 mA. Angular scanning was collected from 5.0 to 80° 2θ with a step size of 0.0260° and rate of 0.6°/min. X-ray photoelectron spectroscopy (XPS) measurements were collected using a VG Scientific ESCALAB MKII (Thermo Scientific, Waltham, MA, USA) spectrometer system using an A1Kα as an excitation source (*hν* = 1486.6 eV). The chamber pressure during analysis was <2 × 10^−8^ mbar. Generally, data acquisition was collected with dwell times of 6 s per data point; to improve signal-to-noise at regions of interest the dwell time per data point was increased to 60 s per point for C1s and O1s, and 200 s per point for Cu2p. After data acquisition the signal of each data point was rescaled to dwell times of 60 s. The spectrum was calibrated by reference to the C1s peak at 284.8 eV binding energy.

The concentrations of copper in the fabric were determined using inductively coupled plasma mass spectrometry (ICP-MS) at the University of Utah ICP-MS Metals Lab. Briefly, 0.500 g of treated fabric was placed in a microwave reactor vessel containing 9.5 mL of 34% nitric acid solution. The reactor vessel was irradiated with microwaves using a microwave reactor (Mars 6 230/60, CEM Corporation, Matthews, NC, USA) for 60 min. The digest was diluted 1:10 w/w with a final internal standard concentration of 10 ppb indium. Concentrations were determined using an external calibration curve using copper single element standard (Inorganic Ventures, Christiansburg, VA, USA).

UV-vis absorbance spectra were collected for the wavelength range of 220 to 800 nm with a step size of 1.0 nm using a UV/Vis/NIR spectrophotometer (ISR-2600, Shimadzu, Tokyo, Japan). Spectral data were analyzed using Origin 2018b Graphing & Analysis software (OriginLab, Northampton, MA, USA).

### 2.5. Time-Dependent Contact Angle Measurements

To determine relative hydrophobicity of cotton samples throughout the synthesis of Cu_2_O NF-cotton, contact angle measurements were obtained for untreated-, Cu(OH)_2−_, [Cu(OH)_4_)]^2−^-, and Cu_2_O NF-cotton fabrics. Onto each textile surface was deposited 5 µL DI water using a Video Contact Angle (VCA) Optima (AST Products, Inc., Billerica, MA, USA). Time-dependent contact angle measurements were collected at automatic time increments of 30 frames per second and calculated using VCA Optima XE software.

### 2.6. Laundering Durability Experiments of Cu_2_O NF-Cotton Fabric

Washing of Cu_2_O NF-cotton fabric (17,489 ppm copper content) was conducted following the AATCC Test Method 61-2007: Colorfastness to Laundering: Accelerated, using a laboratory washing machine, Launder-Ometer (M228-AA, SDL Atlas, LLC, Rock Hill, SC, USA). A stainless steel canister containing 200 mL of 0.37 w/v ionic detergent solution in DI water (Tide^®^, Procter & Gamble Co., Cincinnati, OH, USA) and ten stainless steel balls (6.35 mm diameter) was preheated to 40 ± 0.1 °C in the laboratory washing machine. A treated fabric swatch (50 mm × 200 mm) was added to the preheated canister and rotated at the constant temperature of 40 ± 0.1 °C and constant rate of 40 ± 2 rpm for 45 min. The stainless steel balls and washing conditions for the accelerated laundering procedure is equivalent to five at-home or commercial laundering cycles. The fabric swatches were washed consecutively up to a maximum of 50 cycles. After the desired number of simulated laundering cycles were completed, the canister was removed from the washing machine, and the fabric swatch was rinsed in room temperature DI water for 5 min. and air-dried at room temperature. The percent copper remaining was determined by ICP-MS from acid digestion of Cu_2_O NF-cotton fabrics, and by UV-vis calibration curve and presented as percent copper remaining:(1)$$\left[\mathrm{Cu}\right]\left(\%\right)=\frac{l_{0}-l_{n}}{l_{0}}\times 100,$$
where $l_{0}$ and $l_{n}$ are the concentrations of copper determined by ICP-MS or UV-vis for 0 cycles and *n*th cycles, respectively.

### 2.7. Antibacterial, Antifungal and Antiviral Performance Experiments of Cu_2_O NF-Cotton Fabrics

Antibacterial properties of laundered, Cu_2_O NF-cotton fabric (17,489 ppm copper content) against Gram-negative bacteria *K. pneumoniae* ATCC 4352 and *E. Coli* ATCC 8739, and Gram-positive bacterium *S. aureus* ATCC 6538 were studied following the AATCC test method 100-2007: “Assessment of Antibacterial Finishes on Textile Products” (Microchem Laboratory, Round Rock, TX, USA). Antifungal activity of laundered, Cu_2_O NF-cotton fabric (17,489 ppm copper content) against fungus *A. niger* ATCC 6275 was tested following AATCC test method 30 Test III: “Antifungal Activity, Assessment on Textile Materials: Agar Plate, *Aspergillus niger*” (Microchem Laboratory, Round Rock, TX, USA). Fungal growth on the fabric surfaces were observed under 40× magnification. Antiviral activity of laundered, Cu_2_O NF-cotton fabric against enveloped RNA virus *Human coronavirus, strain 229E*, ATCC VR-740 was determined following AATCC test method 100-2007: “Test Method for Antibacterial Finishes on Textile Materials Modified for Viruses” (Microchem Laboratory, Round Rock, TX, USA). A percent reduction of bacterial growth was calculated using the following equation:(2)$$\mathrm{percent}\;\mathrm{reduction}\;\left(\%\right)=\frac{A-B}{A}\times 100,$$
where A is the number of viable test bacteria on the cotton fabric control sample immediately after inoculation and B is the number of viable test bacteria on the treated cotton fabric test sample after the contact time (24 h).

## 3. Results and Discussion

### 3.1. Internal in Situ Formation of Cu_2_O NFs in Cotton Fibers

The precursor solution of aqueous [Cu(OH)_4_]^2−^ was prepared via a two-step process. First, aqueous NaOH was added to a solution of CuSO_4_ to form Cu(OH)_2_ as a light blue precipitate, Figure 1. Next, addition of excess NaOH, the Cu(OH)_2_ precipitate immediately solubilizes to form a dark blue solution of aqueous [Cu(OH)_4_]^2−^ [38]. To this solution was added a swatch of white cotton print cloth fabric, with UV-vis spectrum and corresponding digital image in Figure 2. Initial studies showed the necessity of excess NaOH, as the UV-vis spectrum had no change upon soaking in neutral pH CuSO_4_ solution. Conversely, the use of aqueous [Cu(OH)_4_]^2−^ and excess NaOH resulted in a visual color change and the increase in absorbance at *λ*_max_ of 290 and 640 nm as a result of the incorporation of Cu^2+^ into the cotton fabric (Figure 2A–C). The UV-vis spectra from exhaustion studies using 0.250 g·100 mL^−1^ and 0.100 g·100 mL^−1^ CuSO_4_ solutions showed decrease in solution intensity and corresponding increase in fabric absorbance over 30 min, Appendix A.

Figure 1 describes the importance of excess sodium hydroxide and the diffusion of ionic [Cu(OH)_4_]^2−^ complexes into the internal structure of the cellulose fibers. Due to the relatively low pK_a_ of cellulose (<10.2) [39,40], the excess NaOH readily deprotonated the hydroxyl groups allowing for [Cu(OH)_4_]^2−^ to be incorporated and form strong coordinate complexes, which have been reported for between cellulose and other transition metals, e.g., Fe^3+^, Ni^2+^, Pd^2+^ [41,42]. With the closely-packed structure of the cellulose chains within the cotton microfibrillar structure, the resulting alkoxides created a hospitable environment for strongly associated 4-coordinate copper complexes that resisted leaching or removal from the fiber. The [Cu(OH)_4_]^2−^ incorporated cotton fiber was rinsed with deionized water removing the excess NaOH and protonating the cellulose alkoxides. This resulted in the precipitation of Cu(OH)_2_ within the cotton fiber with a shift in the *λ*_max_ to 713 nm in agreement with the absorbance of Cu(OH)_2_ (Figure 2A,D) [43]. As previously stated, Cu(OH)_2_ is insoluble in aqueous solution and this property manifests as an imbued hydrophobicity to the cotton material. This was confirmed by measuring the time-dependent contact angle for each of the material with the contact angle measurement quickly decreased to zero for the hydrophilic samples (untreated, [Cu(OH)_4_]^2−^, and Cu_2_O NF) within 0.1 s, and was six times longer for the Cu(OH)_2_ fabric, Appendix A.

The internally deposited Cu(OH)_2_ was thermally stable when the fabric was dried; with no change to the color or UV-vis spectrum when the fabric was heated to 150 °C in an oven for several hours (Figure 2A,D). Alternatively, when the neutralized Cu(OH)_2_–fabric was immersed in a water bath at 80 °C, the fabric color changed immediately from blue to dark brown (Figure 2E). A new peak formed in the UV-vis spectrum for the brown cotton fabric at *λ*_max_ of 320 nm, which corresponded to the surface plasmon resonance of the newly formed reduced oxidation state cuprous oxide nanoflowers (Cu_2_O NF) [44,45]. The thermal stability of the dried Cu(OH)_2_-fabric and the rapid formation of the Cu_2_O NFs in solution demonstrated the necessary role water played in the hydrothermal reduction reaction for producing Cu_2_O NF with cellulose as the reducing agent.

The FE-SEM images and EDS mapping were used for determining the size, morphology and composition of the particles formed on the cotton fibers collected from isolated microfibrils (Figure 3). Figure 3A shows that despite using an 8% NaOH solution to produce Cu_2_O NF-cotton, which is well below mercerization concentrations of 20% the fiber retains its original shape. This is further confirmed based on the XRD spectrum (Figure 4B). The absence of the (020) lattice plane of cellulose II agrees that the crystalline structure of cellulose was not impacted by alkaline treatment. Further magnification of the cotton fiber reveals the formed Cu_2_O NF (Figure 3B). The morphology of the nanoparticles in a nanoflower shape are notable for their high surface-area-to-volume due to irregularity in shape and size, with large aggregates present on the surface of the fibers (Figure 3C). The FE-SEM (Figure 3D) and EDS map of Cu element (Figure 3E) demonstrates the uniform distribution of Cu_2_O NFs on the fabric surface. The EDS spectrum for a point on the surface of the cotton fabrics confirms the presence of Cu element with a strong peak at 0.930 keV corresponding to Cu Lα.

The X-ray diffractograms of the Cu_2_O NF-cotton fabric (Figure 4) showed the characteristic cotton peaks at 2*θ* values 14.7°, 16.6°, 22.7° and 34.8° assigned to the (1–10), (110), (200), and (004) lattice planes of cellulose Iβ, respectively [46,47]. Note, the absence of the (020) lattice plane of cellulose II (Figure 2B) which further indicated that the crystalline structure of cellulose was not impacted by the alkaline treatment. The presence of new peaks at 2*θ* values 29.7°, 36.6°, and 42.4° corresponded to the (110), (111) and (200) lattice planes of Cu_2_O, respectively [24,48], and confirmed that the Cu nanomaterials were successfully produced and immobilized within the cotton fabric.

The X-ray photoelectron spectrum for the Cu_2_O-cotton fabric (Figure 5) exhibited signals at 285 eV and 531 eV corresponding to the C 1s and O 1s regions, respectively of cotton cellulose [49]. Additionally, characteristic Cu 2p_3/2_ and 2p_1/2_ peaks for Cu (I) at 932.2 and 951.8 eV were observed, which agreed with literature assignments [50]. Where copper (II) oxide (CuO) would have had strong, well resolved peaks in nearly the same intensity as the Cu 2p_3/2_ signals around 939 and 942 eV, these are very weak for the XPS spectrum of the Cu_2_O NF-cotton fabrics in Figure 5, which suggested minor oxidation of the surface nanoparticles [51]. The Cu^2+^ ions were incorporated within the closely packed β(1–4) linked D-glucose chain of cellulose and were thus readily reduced and formed Cu_2_O upon neutralization. This is conceptually similar to the Benedict’s test, where cuprous (I) oxide is produced from the reduction of copper (II) in the presence of sugars, and take advantage of the terminal ends of the cotton cellulose chains present in the crystalline and amorphous regions of the fiber, as well as the pseudo-aldehyde structure of the 1–6 bridging ether, as outlined in Figure 1. Previously, cellulose has only been shown to reduce copper (II) to copper (I) oxide under high pressure and temperature conditions using an autoclave [52]. Here, the stabilized Cu^2+^ ions along the deprotonated alkoxide cellulose backbone were reduced under relatively mild, neutral aqueous conditions due to the confined nanoenvironment within the cotton fiber, enhancing the reactivity (Figure 1).

The formation of Cu_2_O NF inside the cotton fiber was confirmed by imaging the copper-fiber nanocomposite cross-section by TEM (Figure 6B), which shows the formation of Cu_2_O NF within the entirety of the microfibrillar structure. Further magnification of the cross-section in Figure 6C,D show the morphology of the Cu_2_O NF in the interior of the cotton fiber are also nanoflowers. These nanocomposites are irregular in shape and size so the Feret diameter—the longest distance possible in any direction of a non-symmetric area—was used to calculate the average diameter. The median and mean sizes for the maximal Feret diameter were 57.8 nm and 72.0 ± 51.8 nm, respectively (Figure 6A).

Figure 7A shows the UV-vis spectra of Cu_2_O NF-cotton with increasing concentration of Cu as determined by ICP-MS by varying the concentration of CuSO_4_ in the reaction solution. With increasing Cu concentration, the UV-vis SPR intensity redshifts from a peak maximum at 242 nm for low concentration (344 ppm) to a peak wavelength of 320 nm corresponding to a significant 17,489 ppm Cu content. With increasing concentration there is greater aggregation of Cu_2_O NFs resulting in redshift. The observed intensity at 320 nm for each concentration was plotted as a function of increasing copper concentration is presented in Figure 7B. The data showed a strong correlation (R^2^ = 0.9979) when fit to an exponential equation. This method was further applied to determine the concentrations of Cu present in laundered fabrics and compared with concentrations determined directly by ICP-MS.

### 3.2. Laundering Durability Experiments of Cu_2_O NF-Cotton Fabric

The Cu_2_O NF-cotton fabric was laundered according to the AATCC test method 61-2007. Copper concentrations were determined by ICP-MS by acid digestion of laundered Cu_2_O NF-cotton fabrics, and by the previously determined calibration curve, for the following laundering cycles; 0, 5, 10, 20, 30, 40, and 50 cycles, with the initial concentration of 17,489 ppm. The percent copper remaining in the fabric for each laundering cycle was calculated using Equation (1). As the number of laundering cycles increased, there was a slight decrease in absorbance at 320 nm. After five launderings the amount of copper released from Cu_2_O NF-cotton fabric was only 9%, and after 50 cycles the fabric only lost 19%. There was strong agreement between the values determined by the previously determined UV-vis calibration curve and ICP-MS, which determines overall copper content irrespective of speciation. The initial drop in copper can be attributed to Cu_2_O NFs on the exterior of the cotton fiber; since the surface-bound Cu_2_O NF were adsorbed electrostatically, the nanoflowers could be detached from the fiber surface through mechanical agitation during the laundering process. The release behavior of copper exhibited a leveling effect upon additional laundering cycles. This was the result of complete removal of the surface-bound nanoflowers, consistent with previous observations for analogous internally dispersed silver nanoparticle cotton fabrics [53].

Examination of the *λ*_max_ across sequential launderings revealed a blueshift in the peak wavelength from 320 nm at 0 cycles to a lower wavelength of 310 nm after ≥5 cycles (Figure 8A). This blueshift in wavelength was attributed to small changes in the surface chemistry of the Cu_2_O nanoflowers during the laundering since there is not a significant difference in copper concentration for 30 to 50 laundering cycles. For example, the surfactants and salts present in the detergent can partially oxidize the nanoflower surface from copper (I) to copper (II), consistent with literature values of CuO *λ*_max_ between 270 and 310nm [54]. Since Cu_2_O NFs are known to exhibit bactericidal and fungicidal properties, the laundered Cu_2_O NF-cotton fabric after 50 cycles was further studied for its antibacterial and antifungal properties against *Staphyloccocus aureus* and *Aspergillus niger*, respectively.

### 3.3. Antibacterial, Antifungal and Antiviral Performance of Cu_2_O NF-Cotton Fabrics

Copper nanomaterials have been long used for their remarkable antimicrobial properties in commercial goods. One limitation to these antimicrobial agents when applied to commercial textiles is reduced efficacy after continuous use and laundering cycles. The Cu_2_O NF-cotton fabric (17,489 ppm copper content) was tested after 50 laundering cycles according to the AATCC test method 100-2007 against Gram-negative bacteria *K. pneumoniae* and *E. coli*, and against Gram-positive *S. aureus*, Table 1 and Figure 9A,B. The Cu_2_O NF-cotton fabric exhibited 99.995% inhibition against Gram-positive bacteria *S. aureus* after just 24 h, with 10^6^ fewer colony forming units compared to control cotton at time zero. Compared to previously reported methods for externally applied Cu_2_O nanoparticles to cotton fabrics that expressed 65% inhibition after 50 laundering cycles [55], and for internally formed cuprous oxide nanoparticles within a thiol-conjugated cotton fabric that exhibited a reduction in antibacterial activity <99% after just 10 laundering cycles in non-ionic detergent [13]. This superior antibacterial activity can be attributed to the internally formed Cu_2_O NFs. Compared to the externally applied Cu_2_O nanoparticles, which are easily detached during laundering resulting in a decreased antibacterial activity, the internally formed Cu_2_O NFs in the cotton fabric are sterically trapped and resist mechanical detachment. These trapped Cu_2_O NFs therefore release bactericidal Cu^+^ ions in a controlled manner. The antiviral activity was also studied for the Cu_2_O NF-cotton fabric against *human coronavirus, 229E strain*, using AATCC test method 100 modified for viruses. The reduction in viral titer was determined to be ≥90% (≥1.00 log_10_), with no virus recovery from the plate even at 10^−1^ dilution of the recovery plate. The antifungal activity of the Cu_2_O NF-cotton fabric was also tested using AATCC test method 30 Test III using fungus *A. niger*. The Cu_2_O NF-cotton fabric after 50 laundering cycles inhibited all fungal growth after 7 days even in a growth promoting environment, Figure 9C,D. This widespread antimicrobial activity emphasizes the application and utility of the Cu_2_O NF-cotton fabric to a multitude of commercial and health-related products.

## 4. Conclusions

Cotton fabrics with multifunctional properties infused with metal oxide nanoflowers for antibacterial applications are becoming increasingly popular [56]. Copper oxide (cupric and cuprous) nanomaterials have gained popularity for their synthetic versatility, microbicidal properties and cost-effectiveness compared to other metals [11,12,13]. This study has presented a method in which cuprous oxide nanoflowers (Cu_2_O NFs) are synthesized in situ from a copper (II) precursor under mild conditions and without added chemical reducing agents. The internal formation of Cu_2_O NFs (72.0 ± 51.8 nm) within the cotton fiber structure was confirmed by TEM. These Cu_2_O NF cotton fabrics demonstrated remarkable laundering durability, releasing only 19% of copper content after 50 home laundering cycles determined by ICP-MS and UV-vis using a calculated calibration curve (R^2^ = 0.9979). The washing durability of the internally produced Cu_2_O NFs was verified by the persistence of superior broad-spectrum antibacterial, antifungal and antiviral activity with greater than 99.99% inhibition against *K. pneumoniae*, *E. coli*, *S. aureus* and *A. niger* and ≥90% inhibition against *Human coronavirus*, *strain 229E*, even after 50 home laundering cycles.

## 5. Patents

The work reported in this manuscript has resulted in U.S. Non-Provisional Patent Application No. 17/371,906 entitled “Cellulosic fibers comprising internally dispersed cuprous oxide nanoparticles”.

## Figures and Tables

**Figure 1 molecules-27-07706-f001:**
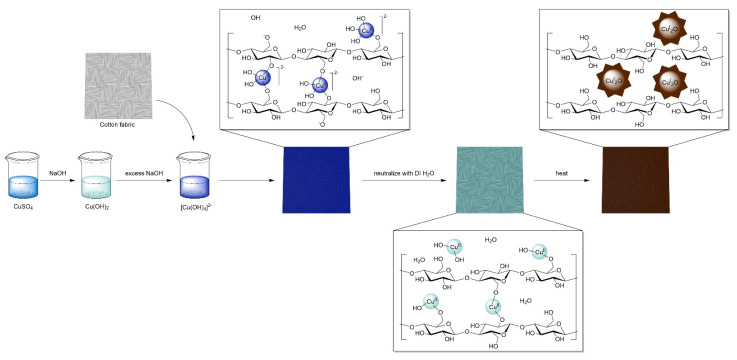
Reaction schematic for the in situ synthesis of Cu_2_O NF-cotton fabrics. Dark blue and light blue spheres represent Cu^II^(OH)_x_ (x = 4 or 2, respectively) metal centers and brown stars labeled Cu^I^_2_O represent Cu*_2_*O nanoflowers.

**Figure 2 molecules-27-07706-f002:**
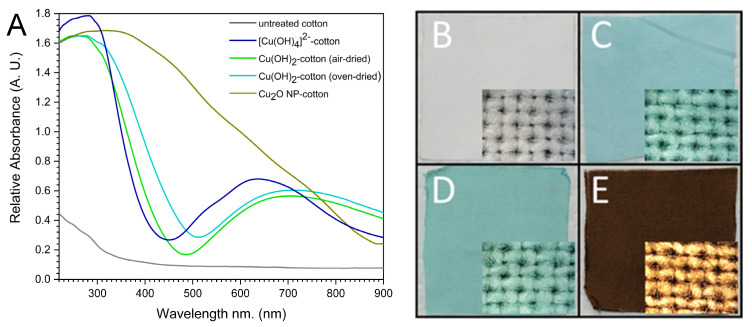
(**A**) UV-vis absorbance spectra and corresponding digital images with digital microscope images inset of (**B**) untreated fabric, (**C**) air dried [Cu(OH)_4_]^2−^-cotton, (**D**) air dried Cu(OH)_2_-cotton, and (**E**) Cu_2_O NF-cotton fabrics.

**Figure 3 molecules-27-07706-f003:**
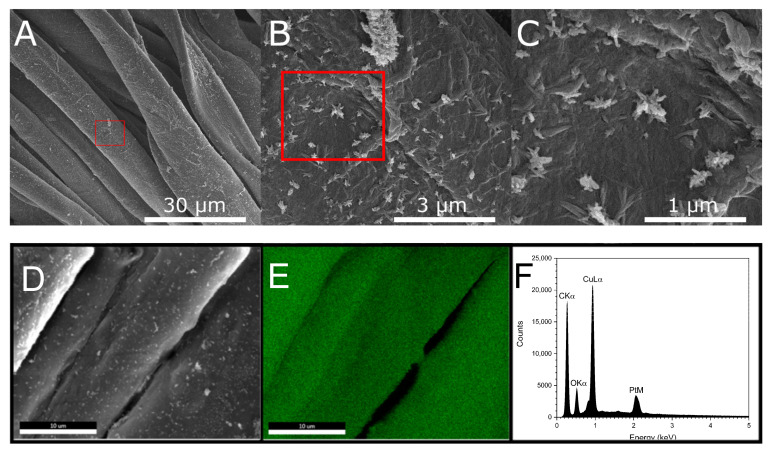
(**A**–**C**) FE-SEM images of Cu_2_O NF produced on the surface of cotton fibers taken with magnifications of 2000×, 20,000×, and 50,000×, respectively. The red outline represents the area of magnification. (**D**) FE-SEM image and (**E**) EDS mapping of the treated cotton fiber for Cu element, and (**F**) corresponding EDS spectrum.

**Figure 4 molecules-27-07706-f004:**
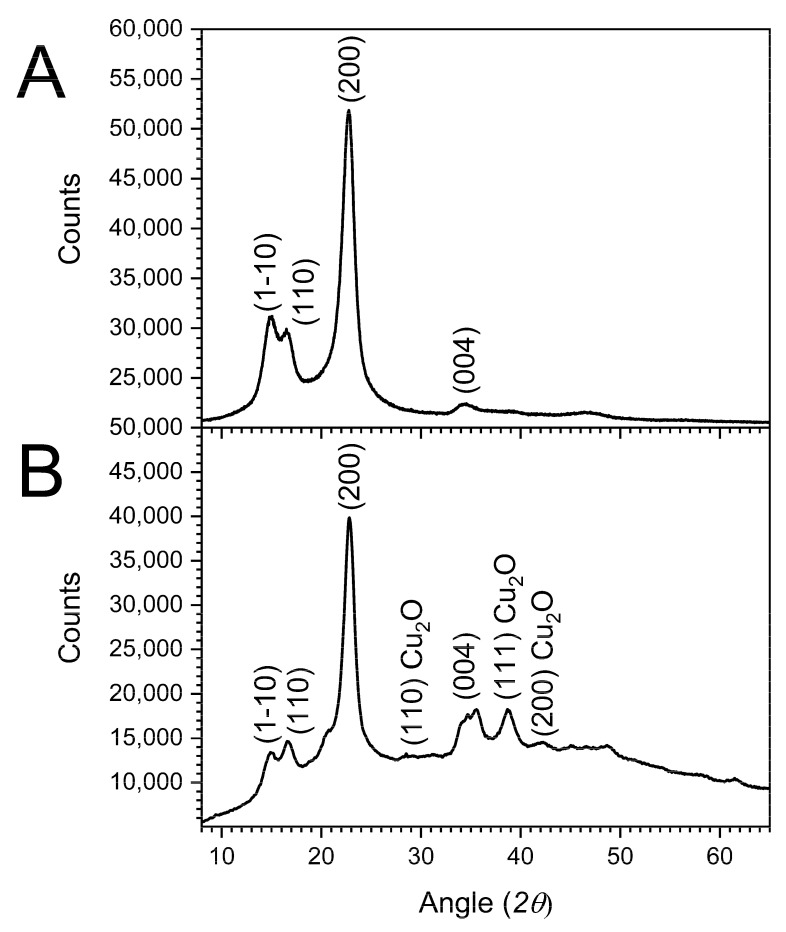
XRD pattern of (**A**) untreated cotton fabric and (**B**) Cu_2_O NF-cotton fabric. Peaks for the cotton cellulose and Cu_2_O NF lattice planes are denoted.

**Figure 5 molecules-27-07706-f005:**
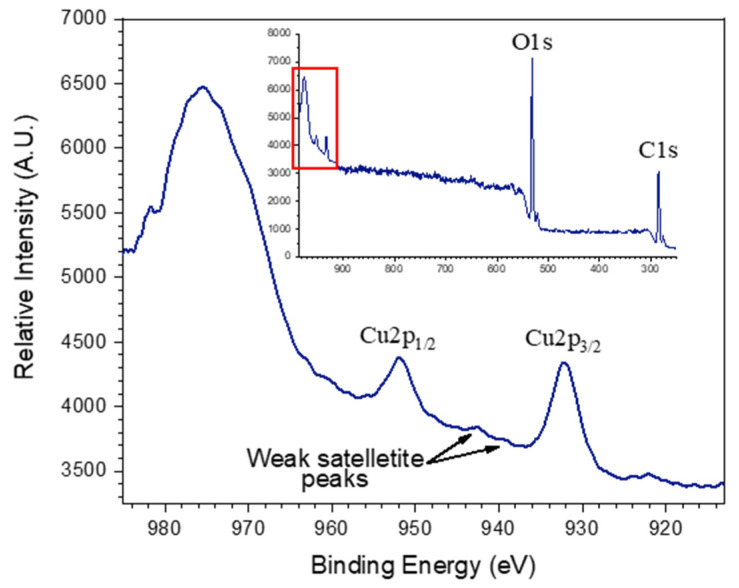
XPS spectrum of Cu_2_O NF-cotton fabric focused on the Cu_2_O peaks and (inset) full spectrum. Peaks of the cotton C1s and O1s are denoted in the inset and Cu2p_3/2_ and Cu2p_1/2_ peaks are denoted in the zoomed spectrum.

**Figure 6 molecules-27-07706-f006:**
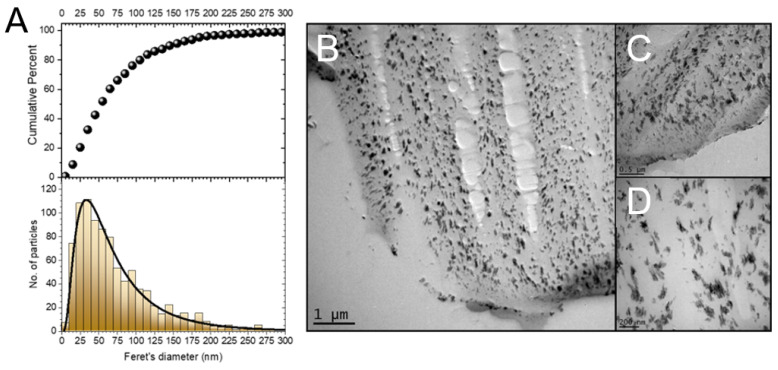
(**A**) Size distributions of Cu_2_O NFs in (**B**) TEM image of the cross-section of Cu_2_O NF-cotton. Cu_2_O NFs were observed across the entire cross-section of the fiber. (**C**,**D**) 2× and 4× magnification of the fiber showing flower-shape of Cu_2_O NFs, respectively. The mean Feret diameter of the Cu_2_O NFs was 72.0 ± 51.8 nm.

**Figure 7 molecules-27-07706-f007:**
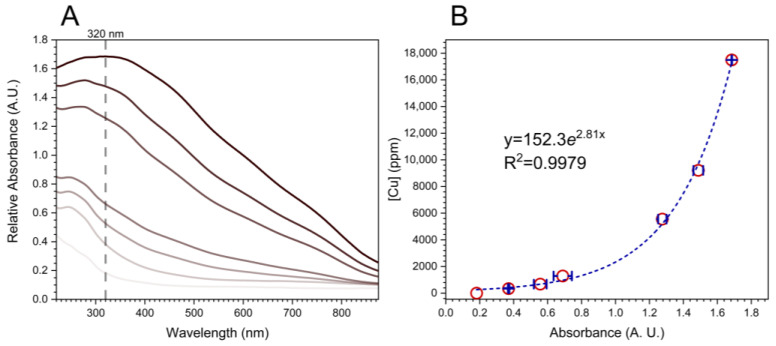
(**A**) UV-vis spectra of increasing copper concentration in Cu_2_O NF-cotton and (**B**) calibration curve of SPR intensity at 320 nm vs. ICP-MS determined copper concentration.

**Figure 8 molecules-27-07706-f008:**
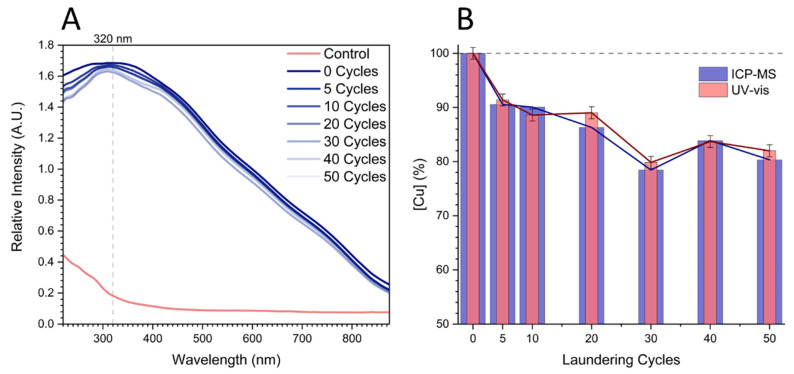
(**A**) UV-vis absorbance spectra of control and Cu_2_O NF-cotton fabrics obtained after incremental home launderings in a detergent solution. (**B**) The percentage copper content remaining in after consecutive laundering cycles of Cu_2_O NF-cotton fabrics as determined by (blue) ICP-MS and (red) UV-vis absorption at 320 nm.

**Figure 9 molecules-27-07706-f009:**
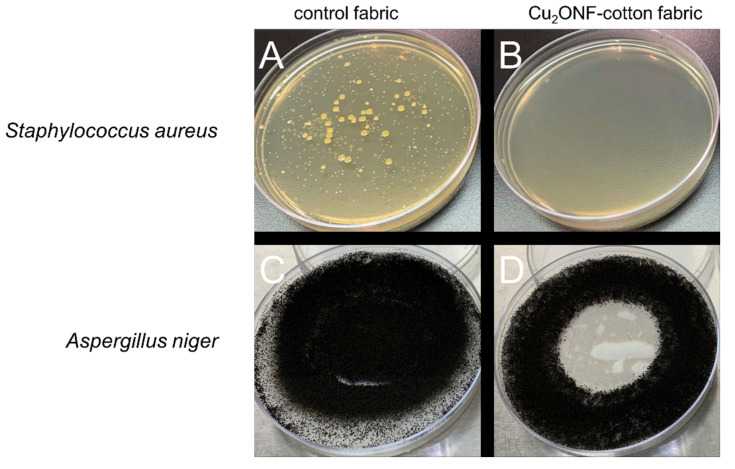
Antimicrobial studies after 24 h for control fabrics and Cu_2_O NF-cotton fabrics for 50 laundering cycles. Antibacterial inhibition against Staphylococcus aureus for (**A**) control and (**B**) laundered Cu_2_O NF-cotton fabrics; and antifungal inhibition against *Aspergillus niger* for (**C**) control, and (**D**) laundered Cu_2_O NF-cotton fabrics.

**Table 1 molecules-27-07706-t001:** Percent inhibition of Cu_2_O NF-cotton fabric against various pathogens.

Pathogen	Classification	Percent Inhibition (%)
*Klebsiella pneumoniae*	Gram-negative bacterium	>99.99994
*Escherichia coli*	Gram-negative bacterium	99.9998
*Staphylococcus aureus*	Gram-positive bacterium	99.995
*Aspergillus niger*	Fungus	Complete inhibition
*Human coronavirus*, *229E strain*	Enveloped RNA virus	≥90

## Data Availability

Not applicable.

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
