# Peer review of "Intrafibrillar Dispersion of Cuprous Oxide (Cu2O) Nanoflowers within Cotton Cellulose Fabrics for Permanent Antibacterial, Antifungal and Antiviral Activity"

_molecules, 2022, doi:10.3390/molecules27227706_

Round 1

Reviewer 1 Report

Comments

In this manuscript, Hillyer and coworkers constructed textiles with in situ synthesis of cuprous oxide nanoparticles on cotton fibers (Cu2O NF-cotton fabrics) by a gentle method that does not require the use of chemical reducing agents. The cotton fabric exhibits high washing durability and can be used for broad-spectrum antibacterial, antifungal and antiviral activities due to the presence of copper oxide. In general, the article is good, but there are some issues that need to be addressed. My suggestions are as follows.

1.     In INTRODUCTION, some previous work in the relevant field may be presented. Please also cite the referenced articles correctly and completely.

2.     Please check all figures, the misplaced serial numbers and pictures are making it difficult to read.

3.     In Figure 2c, the illustration is [Cu(OH)4]2--cotton, and in the article “this corresponds to a transition from the untreated fabric (Figure 2b) to the air-dried (Cu(OH)2) fabric (Figure 2c)” is (Cu(OH)2) fabric.

4.     Figure 6 is incomplete.

5.     The subtitle of “Antibacterial, antifungal and antiviral performance of Cu2O NF-cotton fabrics” is 3.3.

6.     In this article, the antimicrobial capacity after only 50 washes is shown, and there is no comparison showing the repeated use of the material for antimicrobial purposes. Some relevant data can be added.

7.  Some related research about the other antibacterial materials and applications should be cited to compare with the cotton fibers.  e.g., Colloid and Interface Science Communications 28 (2019): 20-28. 

Author Response

  1. In INTRODUCTION, some previous work in the relevant field may be presented. Please also cite the referenced articles correctly and completely.
  2. Please check all figures, the misplaced serial numbers and pictures are making it difficult to read.

Authors response: We apologize for the formatting of the figures. The many issues relating to figure formatting have been corrected in the text.

  1. In Figure 2c, the illustration is [Cu(OH)4]2--cotton, and in the article “this corresponds to a transition from the untreated fabric (Figure 2b) to the air-dried (Cu(OH)2) fabric (Figure 2c)” is (Cu(OH)2) fabric.

Authors response: Thank you for this comment. The disagreement has been remedied in the text.

  1. Figure 6 is incomplete.

Authors response: We apologize for the formatting of the figures. The many issues relating to figure formatting have been corrected in the text.

  1. The subtitle of “Antibacterial, antifungal and antiviral performance of Cu2O NF-cotton fabrics” is 3.3.

Authors response: Thank you for this comment. The error has been remedied in the text.

  1. In this article, the antimicrobial capacity after only 50 washes is shown, and there is no comparison showing the repeated use of the material for antimicrobial purposes. Some relevant data can be added.

Author response: We believe the suggested analysis by the reviewer would be more beneficial and applicable if we were performing wipe studies where repetitive use without laundering would be important to determine. However, here the extensive laundering would have a more significant effect on the antimicrobial performance. Additionally, we employ AATCC test method 100-2007 which uses a contact time of 24 h and a high bacterial load concentration (1.6×106 CFU/Carrier) which should be sufficient to push the limits of antimicrobial activity.

  1. Some related research about the other antibacterial materials and applications should be cited to compare with the cotton fibers.  e.g.,Colloid and Interface Science Communications 28 (2019): 20-28. 

Author response: Thank you for this comment.  References were added.

Author Response

  1. Line 42: an unnecessary “the” in the beginning of the line.
  2. Throughout the text, space between CuxO and NPs is missing. Unless authors specifically want to keep it this way and the journal agrees, please change it. (But then, in Line 77 there is a space)
  3. Line 52: How is cotton damaged by microorganisms and how does that damage lead to negative effects in humans?
  4. Line 66: Please take a line or two how CuxO NPs lead to potential environmental impact.
  5. Line 62: A reference for the claim that most methods require toxic reducing agents and are energy demanding.
  6. Line 64: Ref 26 itself refers to some other papers when making the claim that the authors have borrowed. Perhaps it’s better to cite the original papers instead.
  7. Line 64: Ref 26 says “the methods relying on the synthesis of Cu-based NPs directly on the fiber surface are more often reported” and gives many references. Considering that, what is the novelty that the authors bring to the table through the present study? What is the research gap that the authors want to fill that the other studies have not been able to do?

            Authors response: The previous methods that have been reported on result in the surface application of Cu-based nanoparticles, which have poor durability as a result of the nanoparticles becoming readily detached during use and laundering. Here, we present a method that results in the production of Cu2O nanoparticles within the cotton fiber such that the nanoparticles are nucleated and grown inside the fiber and become too large that they are unable to be liberated. This results in the high durability and retention of nanoparticles during extensive laundering.

  1. Section 2.2: What is the effect of this entire procedure on the cotton fibers? No damage?

Authors response: There is no impact of the synthetic process on the structural integrity of the cotton fibers. We state in the text that the XRD patterns were collected for the untreated cotton and Cu2O NF-cotton. From this, the crystallinity remains intact, which is noted by the well defined (200) lattice planes of cellulose near 22° ().

  1. Is the Equation 2 correct? Should it not be (A-B)/A? If not, please explain.

            Authors response: Thank you for noting this. The error has been corrected in the text.

  1. Number 2 repeated twice for the figures.

Authors response: We apologize for the formatting of the figures. The many issues relating to figure formatting have been corrected in the text.

  1. Figure 2 (but actually Figure 1): Better labelling. Are those star-like things supposed to represent Cu2) NFs?

Authors response: Thank you, this has been corrected in the text.

  1. If the discussion for Figure 2A follows that of Figure 2B/C/D/E, then please change the order of subfigures in Figure 2.

Authors response: Thank you, this has been corrected in the text.

  1. Lines 268, 269, and 270: There is obviously something wrong formatting wise. It’s difficult to follow. Please correct all this.
  2. Line 274: Which figure is it? Where is the label?

Authors response: We apologize for the formatting of the figures. The many issues relating to figure formatting have been corrected in the text.

  1. Where are Figures 6 and 7?

Authors response: We apologize for the formatting of the figures. The many issues relating to figure formatting have been corrected in the text.

  1. Surely there is also degradation from normal wear and tear when the fabric is supposed to be worn. Please comment on it.

Authors response: It is expected that the fiber and fabric integrity will be more impacted by laundering than by normal use [1].

  1. Palme, A.; Idstrom, A.; Nordstierna, L.; Brelid, H. Chemical and ultrastructural changes in cotton cellulose induced by laundering and textile use. Cellulose 2014, 21, 4681-4691, doi:10.1007/s10570-014-0434-9.

Round 2

Reviewer 1 Report

 Accept in present form

Reviewer 2 Report

A study on the fabrication of in situ Cu2O nanoflowers within cotton fibre, and their anti-microbial properties are presented. The manuscript is well written and has been updated to reflect all the suggestions given.